

# corseq: fast and efficient identification of favoured codons from next generation sequencing reads

Salvatore Camiolo and Andrea Porceddu

Dipartimento di Agraria, Università degli studi di Sassari, Sassari, Italy

## ABSTRACT

**Background**. Optimization of transgene expression can be achieved by designing coding sequences with the synonymous codon usage of genes which are highly expressed in the host organism. The identification of the so-called ''favoured codons'' generally requires the access to either the genome or the coding sequences and the availability of expression data.

**Results**. Here we describe corseq, a fast and reliable software for detecting the favoured codons directly from RNAseq data without prior knowledge of genomic sequence or gene annotation. The presented tool allows the inference of codons that are preferentially used in highly expressed genes while estimating the transcripts abundance by a new kmer based approach. corseq is implemented in Python and runs under any operating system. The software requires the Biopython 1.65 library (or later versions) and is available under the 'GNU General Public License version 3' at the project webpage https://sourceforge.net/projects/corseq/files.

**Conclusion**. corseq represents a faster and easy-to-use alternative for the detection of favoured codons in non model organisms.

## INTRODUCTION

The preferential usage of a specific synonymous codons set, also referred to as codon bias, was observed for a variety of organisms as diverse as yeasts, plants, bacteria and metazoan (*Plotkin & Kudla, 2011*). Codons that mirror the most abundant tRNA (known as favoured codons) can speed up the translational machinery and are expected, for this reason, to be preferred by natural selection in highly expressed genes (*Bulmer, 1991*). Knowledge of favoured codons for a host organism is a prerequisite for the design of transcripts that are efficiently translated (*Quax et al., 2015*). Such a goal may be achieved by performing a relative synonymous codon usage analysis of the correspondences (RSCU-COA, as implemented in the widely used software codonw (*Peden, 2000*), http://codonw.sourceforge.net/) . This approach relies on the assumption (not always verified) that codon bias and expression level are strongly correlated. Alternatively, favoured codons can be detected by directly comparing the codon usage of highly and lowly expressed genes (*Chiapello et al., 1998*). This methodology requires both the genic sequences and their expression profile. If a suitable annotation is not available (e.g., for

Corresponding author
Salvatore Camiolo, scamiolo@uniss.it

**Table 1** Expression and sequence data used to identify the optimal codons with the classical expression method.

| Species | RNAseq (SRA database)[a] | Coding sequences[b] |
|---|---|---|
| *Arabidopsis thaliana* | SRR5487682 | Phytozome 11 |
| *Oryza sativa* | SRR5560727 | Phytozome 11 |
| *Escherichia coli* | SRR1796826 | EMBL bacteria |
| *Staphylococcus aureus* | SRR5456207 | EMBL bacteria |
| *Lactobacillus pentosus* | SRR2538344 | EMBL bacteria |
| *Bacillus thuringiensis* | SRR1296010 | EMBL bacteria |
| *Saccharomyces cerevisiae* | SRR5060294 | EMBL fungi |
| *Candida albicans* | SRR5444919 | EMBL fungi |
| *Fusarium graminearum* | SRR2170110 | EMBL fungi |
| *Mus musculus* | SRR5252256 | EMBL metazoan |
| *Drosophila melanogaster* | SRR3659025 | EMBL metazoan |
| *Caenorhabditis elegans* | SRR6792646–SRR6792648 | EMBL metazoan |

**Notes.**
[a]The NCBI SRA database is available at http://www.ncbi.nlm.nih.gov/sra/.
[b]Phytozome 11 is available at http://genome.jgi.doe.gov/pages/dynamicOrganismDownload.jsf?organism=Phytozome, EMBL databases are available at http://fungi.ensembl.org/index.html, http://bacteria.ensembl.org/index.html, http://metazoa.ensembl.org/index.html.

a new non-sequenced organism) transcripts may be reconstructed by mapping RNAseq sequence data on a reference genome or by direct *de novo* assembly, both approaches representing a time consuming and hardware demanding task (*Conesa et al., 2016*). Here we propose corseq, a user friendly, fast and reliable tool for the direct identification of the favoured codons from RNAseq data without prior knowledge of genomic/genic sequence or gene expression data. The presented software aims to identify the putative open reading frame (ORF) within the sequences generated by an RNAseq experiment and quantify the relative codons frequency. corseq uses a nucleotide kmer analysis to infer those codons that are preferentially used in highly abundant transcripts by considering the kmer frequency as a proxy of the expression value.

# MATERIAL AND METHODS

## Sequences and expression data

Coding sequences for the two plants *Arabidopsis thaliana* and *Oryza sativa* were downloaded by the Phytozome 11 database; for the four bacteria *Escherichia coli*, *Staphylococcus aureus*, *Lactobacillus pentosus*, *Bacillus thuringiensis*, sequences were obtained from the EMBL bacteria database; for the two fungi *Saccharomyces cerevisiae* and *Fusarium graminearum*, sequences were downloaded from the EMBL fungi database; and for the three metazoan *Mus musculus*, *Drosophila melanogaster*, *Caenorhabditis elegans*, sequences were obtained from the EMBL metazoan database (see Table 1). For the same species, RNAseq data were downloaded from the NCBI SRA database (see Table 1 for the corresponding accession numbers).

## Software implementation

corseq is implemented in Python and can run under any operating system. The software (available at https://sourceforge.net/projects/corseq/files) does not necessitate any

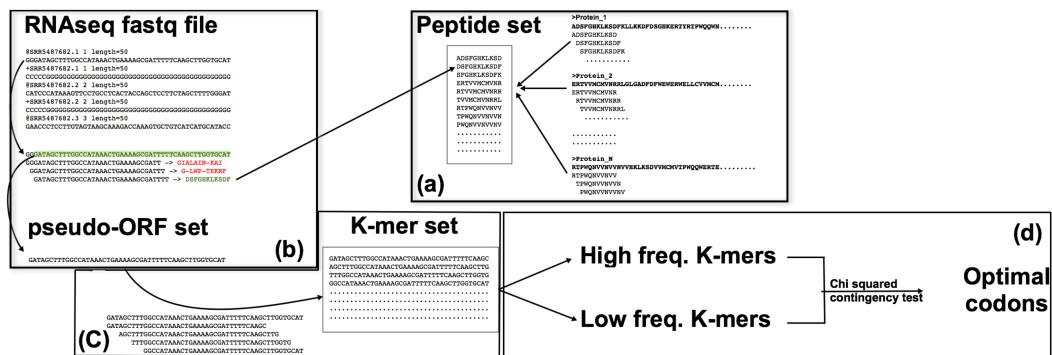

**Figure 1** **The corseq pipeline.** (A) Population of the "peptide set"; selected proteins (suitable for the analysed species) are subjected to a k-mer analysis using a sliding window approach, thus producing a set of amino acid kmers. (B) Identification of the pseudo-ORFs; RNAseq reads are sampled at each position by using a sliding window approach; the produced subsequences are translated and, if stop codons are not present, the corresponding pseudo protein is compared with the set of previously identified amino acid kmers. (C) Quantification of the nucleotide k-mers generated by the pseudo-ORFs; sequences for which the corresponding protein matched an amino acid k-mer, are split in their k-mers that are thus quantified. (D) Favoured codons computation; high and low frequency kmers are considered as proxy of high and low expression genes respectively; favoured codons are then computed by comparing, for each amino acid, the usage of the synonymous codons between these two sequences datasets.

installation and only requires the library Biopython 1.65 (*Cock et al., 2009*) or later version. Corseq takes in input an RNAseq fastq formatted file together with a multifasta protein training set. The software already includes general training sets for viridiplantae, fungi, bacteria and metazoan although user defined proteins can be added. Provided training sets comprise the UniProt proteins (http://www.uniprot.org) filtered for the kingdoms viridiplantae (38,739 sequences), fungi (32,674 sequences), bacteria (333,047 sequences) and metazoan (105,209 sequences). The corseq pipeline starts with the computation of all the amino acids kmers present in the selected protein training set (the flag kp allows the selection of the kmer size at this stage). The identified sequences are stored into a "Peptide set" (Fig. 1). Here, all the presented results involve the use of 11 amino acids kmers, but this parameter can be changed by the user. From the selected RNAseq file, a user-defined number of bases (NOB) are extracted and used to perform the downstream analysis. To do so, the needed number of reads is randomly selected from the original dataset. Each of the selected reads is then searched in both forward and reverse orientations for nucleotides 33-mers (e.g., 3*kp) that codes for any of the 11-mer amino acid sequences present in the Peptide set. When a match is found, the corresponding position and orientation are considered indicative of the original transcript correct frame. The reads are thus cut accordingly to the found position/orientation and stored in what we called as a pseudo-ORF set (Fig. 1). For each of the found pseudo-ORFs, the enclosed nucleotide kmers are counted by performing a sliding window analysis toward the sequence with a window size equal to user defined value (flag −k in the corseq options) and a window step equal to 3. Extremely recurrent kmers are considered as portions of highly expressed genes and, *vice versa*, kmer with few occurrences are more likely to derive from lowly

expressed genes. All kmers are expected to contain (or to be enriched in) in frame codons sequences due to the method used for the generation of the pseudo-ORF and to the selected window step = 3. After sorting the kmers in ascending frequency order, the 2% most frequent and the 2% less frequent sequences are selected. A chi-squared contingency test analysis is then performed for each codon between these two groups. For each codon, $2 \times 2$ contingency tables are filled according the following scheme: the first row reports the number of occurrences of the codon and the number of occurrences of its synonymous peers in high frequency kmers while the second row reports the same values found in the low frequency kmers (*Zhou, Weems & Wilke, 2009*). An odds ratio and a significance value are generated so that odds ratio >1 with a significance level >3.84 ($p < 0.05$) are considered indicative of optimality (see File S1 for a more detailed description of the algorithm and the user definable parameters). The effect of the kmer frequency percentage threshold (corseq default value = 2%) was investigated and consistent results were obtained when this value ranged between 1% and 15% (File S2).

## Conventional expression method

For each analysed species, an RNAseq experiment together with transcripts and coding sequences was downloaded from publicly available databases (Table 1). Favoured codons were first identified by comparison of codon usage profiles between the most extremely highly and lowly expressed coding sequences, an approach that has proven to be an effective means to reveal preferences in codon use that are linked to translation (*Shields et al., 1988*; *Duret & Mouchiroud, 1999*; *Ingvarsson, 2008*; *Cutter, 2008*; *Wang et al., 2011*; *Whittle & Extavour, 2015*). Hereafter we refer to this method as the conventional expression method (CEM). Briefly, RNAseq sequences were aligned to the coding sequences with bwa (*Li & Durbin, 2010*) (default parameters) and transcript abundance were quantified by the tool express (*Roberts & Pachter, 2013*) (default parameters). We then used a chi-square test to compare the codon usage of the 5% most highly and most lowly expressed transcripts, i.e., a method that is implemented in the software Seforta (*Camiolo et al., 2014*). At this stage, $2 \times 2$ contingency tables were created similar to those discussed in the previous section.

## RESULTS

### Result consistency among replicates

Since corseq randomly selects a user defined number of bases (NOB) from the original RNAseq dataset to run the analysis on, we wondered whether the identified favoured codons were consistent when repeating the analysis several times. Moreover, we checked whether possible fluctuations were associated to the selected number of bases. For this reason we carried out five replicates of each of the following NOB values: 5 Mb, 10 Mb, 20 Mb and 40 Mb. Such a test was performed on *A. thaliana*, *S. cerevisiae*, *E. coli*, *M. musculus*, and *D. melanogaster*. Our results show only few non-concordant favoured codons with this trend being further minimized, as expected, for higher NOB values (Fig. 2). For this reason, the corseq default value for the selected number of bases is set to 20 millions.
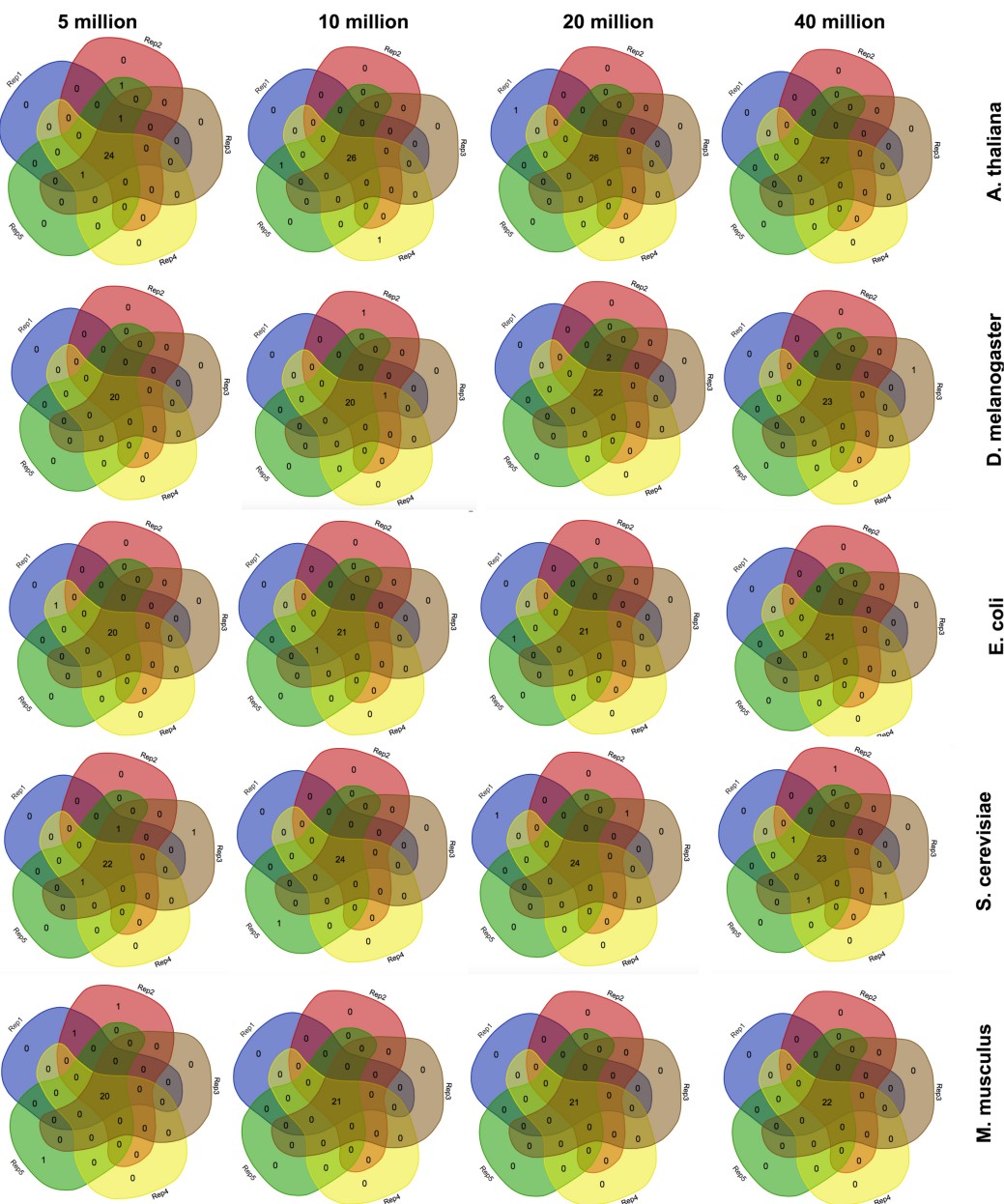

**Figure 2** **Consistency among replicates.** Venn diagrams showing the number of detected favoured codons for five replicates, four values of number of bases and five species. For each presented diagram, each colour region reports the number of codons that were found to be favoured following a corseq run. Overlapping regions reports the number of concordant favoured codons between five replicates for the specified combination of species/number of bases.

## The effect of the nucleotide kmer size

Estimation of highly and lowly expressed genes is performed by computing high and low representative k-mers in the pseudo-ORFs (see 'Material and Methods' section). We tested the effect of the k-mer size in detecting the favoured codons. Hence, we performed five

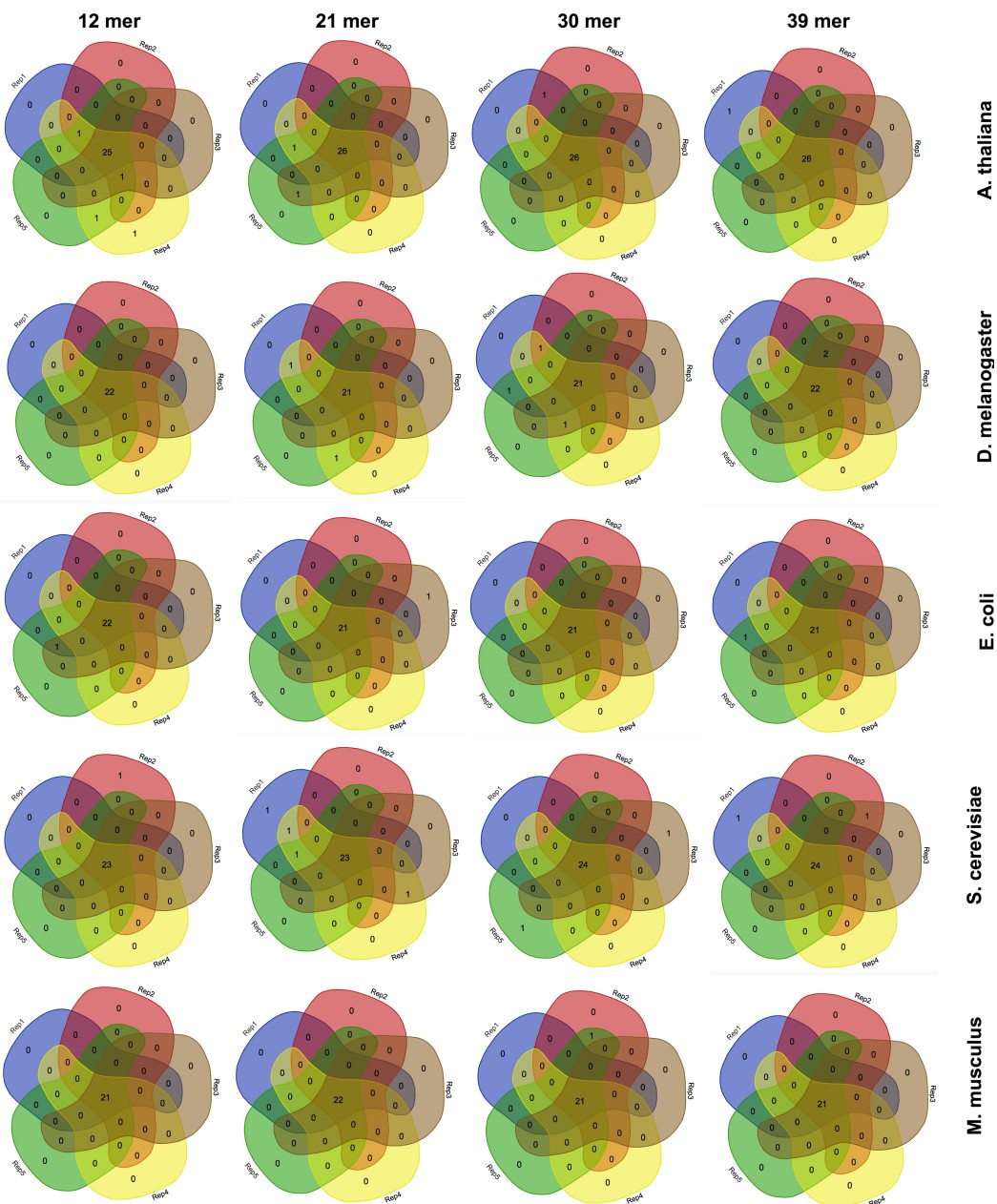

**Figure 3** **The effect of the k-mer size in the detection of favoured codons.** Venn diagrams showing the number of detected favoured codons for five replicates, four nucleotide k-mer sizes and five species. For each presented diagram, each colour region reports the number of codons that were found to be favoured following a corseq run. Overlapping regions reports the number of concordant favoured codons between five replicates for the specified combination of species/nucleotide k-mer size.

replicates for 4 k-mer sizes (12 mer, 21 mer, 30 mer, 39 mer) and for 5 species (*A. thaliana*, *S. cerevisiae*, *E. coli*, *M. musculus*, and *D. melanogaster*). As reported in Fig. 3 few differences were observed among the tested k-mer sizes. However, since lower values may possibly be associated to an increased chance that some k-mer is shared between highly and lowly

**Figure 4 The effect of the kmer size on the corseq/CEM comparison.** Venn diagrams showing the number of detected favoured codons that are shared between corseq with four different k-mer sizes and the conventional expression method. For each presented diagram, each colour region reports the number of codons that were found to be favoured following a corseq run. Overlapping regions reports the number of concordant favoured codons between four replicates and the conventional expression method for the specified species.

expressed genes we decided to leave 39 mer as the default k-mer size for corseq. This value also proved to slightly improve the number of favoured codons that are called concordantly with the conventional expression method (CEM, Fig. 4).

## The effect of the provided number of reads

The software performance, as compared to the CEM, was estimated by using a large RNAseq dataset (9,955,999 reads, sequence length = 150 nt). We wondered how the detection of the favoured codons may be influenced by reducing the initial dataset size. Thus, we randomly selected a number of reads from the *Arabidopsis thaliana* RNAseq file ranging from as low as 25,000 reads (accounting for 3,75 Mb) up to a maximum of 2,000,000 reads (accounting for 300 Mb). We then ran corseq by collecting 20 Mb with a nucleotide k-mer size of 39. Notably, although 25,000 reads are not sufficient to fulfil the requested 20 million bases, a suitable dataset is nevertheless created since extraction of random reads is performed with replacement (e.g., the final dataset is more likely populated by reads deriving from the most abundant transcripts). The performance was estimated by calculating a Pearson correlation coefficient between the odds ratio of the codons calculated by both corseq and CEM with the latter being carried out on the original RNAseq dataset. As expected, the calculated correlation increased at higher dataset sizes up to the value of 125,000 reads (accounting for 18,75 Mb) when a plateau was reached. Notably, even for the smallest RNAseq dataset the odds ratio correlation proved to be higher than 0.8 (Fig. 5A). Size of the analysed RNAseq file represents the primary determinant of the run times. In this regard, corseq was able to run the described analyses in a time range comprised between 2.1 and 12.9 min (Fig. 5B) while the analysis on the entire RNAseq file took as low as 42.5 min to complete.

## Performance benchmark

Favoured codons were calculated by using corseq with default parameters (kmer size for the proteins training set = 11, nucleotide k-mer size for the pseudo-ORF = 39, number of collected bases = 20 millions) for the 12 species reported in Table 1. The same analysis was performed with the CEM as described in the main manuscript. The two outcomes were compared by considering the following parameters: (a) the number of concordant
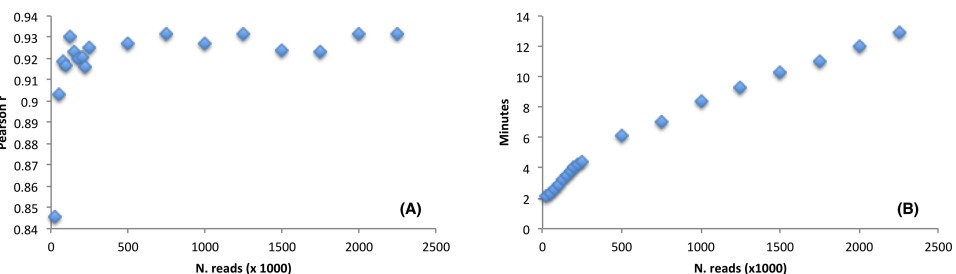

**Figure 5** **The effect of the initial dataset size on the favoured codons detection and running time.** (A) Pearson correlation between the codons odds ratio calculated with corseq and the classical expression method. For each point, the corseq odds ratio values were calculated starting from RNAseq dataset of different size. (B) Computational times of the corseq elaboration for RNAseq dataset of different size.

**Table 2** **Corseq benchmark results.** The table reports the number of optimal codons that are commonly detected by both corseq and the CEM, the number of optimal codons only detected by corseq, those only detected only by CEM, the percentage of optimal codons sharing the same odds ratio direction (e.g., either >1 or <1) in the two methods, the Pearson correlation $r$ between the odds ratio calculated with the two methods (all correlations are significant with $p < 0.05$) and the percentage of amino acid consensus (AAC) that is the ratio of amino acids for which at least one of the identified optimal codons is shared between the two methods.

| | Common | corseq specific | CEM specific | Same odd ratio direction (%) | r | AAC (%) |
|---|---|---|---|---|---|---|
| *A. thaliana* | 21 | 6 | 4 | 0.86 | 0.93 | 0.89 |
| *O. sativa* | 23 | 2 | 3 | 0.92 | 0.95 | 1 |
| *E. coli* | 21 | 1 | 0 | 0.98 | 0.99 | 0.94 |
| *S. aureus* | 14 | 4 | 4 | 0.90 | 0.72 | 0.76 |
| *L. pentosus* | 15 | 5 | 6 | 0.86 | 0.82 | 0.88 |
| *B. thuringenesis* | 16 | 3 | 2 | 0.93 | 0.87 | 1.00 |
| *S. cerevisiae* | 21 | 0 | 2 | 0.97 | 0.9 | 1 |
| *F. graminearum* | 17 | 4 | 8 | 0.78 | 0.72 | 0.83 |
| *C. albicans* | 20 | 1 | 3 | 0.97 | 0.94 | 0.94 |
| *M. musculus* | 21 | 4 | 5 | 0.83 | 0.72 | 1 |
| *D. melanogaster* | 20 | 4 | 2 | 0.95 | 0.92 | 1 |
| *C. elegans* | 20 | 3 | 1 | 0.98 | 0.75 | 0.89 |

favoured codons, (b) the number of method specific favoured codons; (c) the percentage of codons for which the odds ratio featured the same direction with the two methods (odds ratio >1 for both or odds ratio <1 for both); (d) the Pearson correlation between the odds ratio calculated with the two methods; (e) the amino acid consensus (AAC) that is the percentage of amino acids for which at least one of the called favoured codon (e.g., more than one synonymous codon may be called as favourite) is concordant between the two methods. The results are reported in Table 2.

## DISCUSSION

Optimizing the coding sequence of a heterologues transcript is the first step to maximize its expression in a host organism. This process is primarily performed by designing a

coding sequence featuring the same codons that are used in a host's highly expressed genes, e.g., its favoured codons. Notably, such an approach does not rule out the importance of sub-optimal (or even rare) codons in increasing the expression of a heterologues gene (*Shah & Gilchrist, 2011*; *Porceddu, Zenoni & Camiolo, 2013*; *Mauro & Chappell, 2014*; *Gilchrist et al., 2015*; *Camiolo, Sablok & Porceddu, 2017*), since it is known that additional determinants may play a primary role as, for example, the protein co-translational folding (*Yang, Chen & Zhang, 2014*) and the mRNA stability (*Wu et al., 2004*).

Here we present corseq, a new bioinformatics tools capable of quickly computing the favoured codons of an organism by analysing RNAseq data. The software allows for the determination of the codons that are used in the highly expressed transcripts but does not imply an actual quantification of the gene expression. Instead, corseq aims to identify position and orientation of in frame exons (or portions of them) within the RNAseq reads and computes the codons that are more likely to occur in the most abundant transcripts as a result of a combination of k-mer analysis and sliding window approach (see 'Material and Methods' section). First, corseq randomly samples the provided RNAseq fastq file in order to work only on a data subset and thus reduce the run time. Our tests proved that sampling as low as 20 millions bases allows to minimize the risk of calling different favoured codons sets among replicates (Fig. 2). This number also ensures the maximum concordance with the favoured codons found with alternative methods (e.g., the conventional expression method, Fig. 5). Then, the software extracts from each read a pseudo-ORF by inferring its position and orientation from the homology of their coded peptides as compared to a reference protein database (Fig. 1). From each pseudo-ORF all possible kmers are computed with the most recurrent being considered as belonging to highly expressed genes. The effect of the k-mer size was investigated and a good agreement among all the analysed values was found (Fig. 4).

We benchmarked corseq versus one of the most common methods for the identification of favoured codons, herein referred to as the conventional expression method (see 'Material and Methods' section). While a good agreement was found between the two approaches (Table 2), few differences emerged that deserve a careful investigation. Indeed, codons that are found only by corseq should not be necessarily treated as false positive. In fact, corseq generally achieves higher significance levels (due to the increased number of comparisons) as compared to the CEM which may increase its sensitivity in calling as favoured those codons that are only slightly over used in highly expressed genes or whose coded amino acid is poorly represented in the proteome. As an example, in *Arabidopsis thaliana* the cysteine codon TGC is called as favoured by corseq but not by the CEM. This may be possibly due to the cysteine being one of the less frequent amino acids in plants and the identification of its favoured codons can be difficult if a higher number of genes is not analysed. In the effort to understand whether TGC is a false positive or, on the contrary, is the result of the high corseq sensitivity we investigated the genomic tRNA database (*Chan & Lowe, 2016*) and found that the anticodon for TGC is indeed the most abundant. Similarly, in Arabidopsis ACC test failed to find a concordance in the called favoured codons for the amino acid Proline with corseq calling CCA and the CEM calling CCT with the former actually featuring the highest number of anticodon tRNA in this plant species.

This phenomenon was observed for several other codons in several other species (e.g., TGC and GAC in *L. pentosus*, GAG in *B. thuringensis*, CAA in *C. elegans*).

All the odds ratio and significance values for the comparisons described in this benchmark are reported in File S3.

## CONCLUSIONS

The described software is a valuable tool for the identification of the favoured codons for organisms with neither genomic nor annotation data available. We tested corseq on several organisms spanning from plants to bacteria, fungi and metazoan and found, overall, favoured codon sets featuring high agreement with those calculated by the conventional expression method. Indeed, a 0.85 average correlation was observed between the chi-squared odds ratio found with the two methods whereas the percentage of favoured codons detected by corseq and concordant with the CEM ranged from 75 to 100%. Finally, most of the favoured codons identified by corseq were also previously detected by the mean of alternative methods in *S. cerevisiae*, *E. coli*, *M. musculus*, *D. melanogaster* (*Zhou, Weems & Wilke, 2009*), *A. thaliana*, *O. sativa* (*Camiolo, Sablok & Porceddu, 2017*), *S. aureus* (http://codonw.sourceforge.net/JohnPedenThesisPressOpt_water.pdf), *C. albicans* (*Lloyd & Sharp, 1992*) and *C. elegans* (*Riddle et al., 1997*).

Corseq is fast (in our tests the running time ranged from few minutes to around three hours for the largest fastq file), reliable and, to the best of our knowledge, it represents the only alternative to the much more complex methodology involving transcript *de novo* assembly/annotation/reads alignment/expression quantification/favoured codon computation.

## ACKNOWLEDGEMENTS

We would like to thank Dr. Laura Mula and Dr. Emanuela Spanu for their useful suggestions at the initial stage of the software development.

### Funding

The authors received no funding for this work.

### Competing Interests
The authors declare there are no competing interests.

### Author Contributions
- Salvatore Camiolo conceived and designed the experiments, performed the experiments, analyzed the data, prepared figures and/or tables, authored or reviewed drafts of the paper, approved the final draft.
- Andrea Porceddu conceived and designed the experiments, analyzed the data, contributed reagents/materials/analysis tools, authored or reviewed drafts of the paper, approved the final draft.

## Data Availability

Sourceforge: https://sourceforge.net/projects/corseq/files.

## Supplemental Information

Supplemental information for this article can be found online at http://dx.doi.org/10.7717/peerj.5099#supplemental-information.

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
