# Peer review of "corseq: fast and efficient identification of favoured codons from next generation sequencing reads"

_PeerJ, doi:10.7717/peerj.5099_

## Round 0.1 · original submission · Major Revisions

Dear authors,

Thanks for your submission. As you can see from these three reviews there are opportunities to improve the clarity, English and rationale of the submitted manuscript. For example, at least 2 reviewers wondered why 2% was chosen. All three reviewers wanted details on why certain things were chosen/accomplished and, in particular, why Venn diagrams are used.

Also please refer to Mike Gilchrist's public review re: mutation bias and also the original "Codon Adaptation Index" paper(s). In short, only highly expressed genes tend to matter (but I haven't read this myself given I know two reviewers well). The line plots in the 2011 PNAS paper show how things change with expression and how some tools (our own % min/max included) use the average values as a proxy but that might not be ideal.

Hopefully these comments help you prepare and improve the readability of this work as all three seem to find the big picture interesting.

Best,
- S

Reviewer 1 ·

Basic reporting

1. A more thorough overview of what you are trying to do and how corseq accomplishes that (similar to what is in the discussion) placed before the "software implementation" section would be helpful for the reader
2. You focus on "optimizing" codons. It might be worth noting that there are various other studies claiming that sub-optimal codons at specific positions are important in protein formation to facilitate co-translational folding.
3. Figure 1's layout is a little confusing
4. Some minor grammatical/language issues throughout

Experimental design

Why do you use only the top and bottom 2% of kmers? Why not 1% or 3%? Add some reasoning here in the "software implementation" section.

Validity of the findings

No Comment

Additional comments

I think there might be a minor error in "The effect of the provided number of reads" section. Based on your other calculations your reads are 150bp, meaning that the bp coverage from 25,000 reads should be 3,750,000bp and not 3,500,000bp.

Reviewer 2 ·

Basic reporting

The paper presents a method to identify codons that are preferably used in highly expressed genes in a given organism, given as input RNA sequence fragments and amino acid sequences of proteins from this organism.

The paper is generally clear, except for its methods section, which needs additional information. It is not clear how the chi square test is performed, on which values, and how the results are interpreted. Equations would help to document the methods used. Currently it is not possible to reproduce the research with the information provided. The "conventional expression method (CEM)" is not described in sufficiently detail, and references to tools are used to substitute to descriptions of the method itself. It is also unclear to me why the Pearson correlation coefficient value is used instead of the corresponding p-value, which would be significantly more informative.

English could be improved with some proofreading, to catch small errors like "were" instead of "was" on line 118, the use of the word "elaboration" throughout the manuscript, etc.

References and background are sufficient and support the premise of the paper, as well as the objectives. The structure, figures, and tables of the article seem sufficiently professional. I could not find though any raw data, the only supplemental material I could locate was a manual for the corseq software. The article is mostly self contained, except for the methods' descriptions, as mentioned above. The results are relevant to the hypothesis.

Experimental design

In my opinion the primary research of this article falls within the aims and scope of the journal. The research question is well defined and meaningful, and solving the stated problem would be useful to researchers in the field of heterologous expression.

The proposed solution is interesting, but the technical part is dominated by empiricism instead of analytical methods. K-mer size for example is determined by experimenting with seemingly random values of k, settling on the largest one that seems to reduce the number of k-mers shared between over- and under-expressed genes. Why not a higher value then? And why the 2% of the most frequent and least (not "less") frequent sequences are selected from the k-mers? How is this 2% value determined? All this information would help determine better the technical soundness of the methodology used. As presented currently, it is hard if not impossible to replicate the findings.

Validity of the findings

The findings of this paper are interesting and the conclusion is well stated. It may still be difficult to determine and even further quantify the effect this method's preferential codon selections would have on gene expression vs other methods, but that is probably outside the scope of this research.

·

Basic reporting

See attached

Experimental design

See Attached

Validity of the findings

See attached

Additional comments

See attached

---

## Round 0.2 · Minor Revisions

Please see the reviews attached. Since these are largely minor revisions and both reviewers feel the paper is much approved, I will be able to do the final review myself (i.e., no need for additional reviews). Please see Gilchrist's discussion of "adaptive" vs. optimal as this is a valid point given the diversity of codon literature out there that have different underlying models.

Reviewer 1 ·

Basic reporting

My main critique is a few very minor grammatical errors that remain:
line 31: "to infer the" to "the inference of"
line 193: "design" to "designing"
line 202: I think you mean something like "for the determination of" instead of "to investigate"
line 222: "In facts" to "In fact"

Experimental design

No comment

Validity of the findings

No comment

Additional comments

The paper was much more clear and readable following the revisions. My concerns from the first round of review were addressed. The only revisions I would recommend are the few minor grammar issues noted above.

·

Basic reporting

While the authors should be commended for addressing my previous comments and those of the other
reviewers, I do have one major concern, which should be easy to address, and a number of more minor
corrections that prevent me from being able to recommend this paper from being published in its current
form.
As a result of the authors responses and my re-reading of the paper, I have a fundamental problem
with their use of the term ’optimal’. In the codon usage world, ’optimal’ usually refers to the one codon
out of a family of synonyms most favored by natural selection. Here the authors apply the term to any
codon whose usage increases between the low and high expression sets. This is inconsistent with previous
usage and inaccurate. The authors’ comparisons allow them to identify ’selectively favored’ or ’adaptive’
codons and not necessarily all of them since as we illustrate and discuss in our work the authors now cite
(e.g. Shah and Gilchrist 2011 and Gilchrist et al 2015), for amino acids with 3 or more codons, we can see
non-linear changes in a codon’s frequency. Specifically, the frequency of a codon can actually increase and
then decrease with protein production rate. For example, in Fig 6 from Gilchrist et al (2015, attached), we
can see how the frequency of Ile’s ATT initially increases and then decreases with protein production rate,
φ. This is because ATT is selectively favored over the pessimal codon ATA and mutationally favored over
the optimal codon ATC. As a result, ATT frequency can initially increase with φ because it is more fit than
ATA which is initially used with non-zero frequencies. In turn, both ATT and ATC frequency increases with
φ at the expense of ATA. However, once gene expression gets moderately high (around ln(φ) = 0.75, ATA
is so selectively disadvantageous that it is no longer used and only ATT and ATC are competing with one
another. As a result, when increasing ln(φ) above 0.75, we see ATC frequency increase at the expense of
ATT. If selection was efficacious enough and/or protein translation of the genes with the highest rates, we’d
only see ATC, but that’s not the case in yeast and not likely the case for codon families with 3 or more
members other organisms. Thus, I’m going to insist that the authors use a different term than ’optimal’,
such as ’selectively advantageous’ or ’favored’ or ’adaptive’.
In a future study, the authors might want to see how well our code works with the k-mer data they are
using. I suspect if they can get 50-100 codons for about 1000 genes, they might be able to estimate our
model parameters ∆M and ∆η, especially if they can use the density of reads per gene per codon as proxies
for φ.


Minor Criticisms
1. The language is overall good, but there are numerous typos and inconsistencies. Here are a few
examples from the .docx file with tracked changes
(a) line (l) 113: vice versa is two words. (as is de novo l 69)
(b) pseudo-Orf (l 222) vs pseudo-ORF (l 158).
(c) l 83 lists only two, not three, fungal species.
(d) Not sure what the authors mean by ’elaboration time’ on line 183.
2. The significance column Supplemental File 3 is not defined. Are the numbers − log10 (p)?
3. I still think the Venn diagrams are essentially uninformative. Why not just have a graph with how the
frequency of concordance changes with sample size?

Experimental design

No Comment

Validity of the findings

No Comment

Additional comments

No comment

---

## Round 0.3 · accepted · Accept

Thanks for the point-by-point response and tracked changes. I am personally not sold on the Venn diagrams either but you have a sound reason and you understand the data better than we do. Best, - S

#